# 'Intraoperative predictors for clinical outcomes after microinvasive glaucoma surgery"

**Aparna Rao** [ORCID]\*, **Sujoy Mukherjee**

Glaucoma Service, LV Prasad Eye Institute, Bhubaneswar, India

\* aparna@lvpei.org, vinodini10375@yahoo.com

**Data Availability Statement:** All relevant data are within the paper and its Supporting Information files.

**Funding:** The author(s) received no specific funding for this work.

## Abstract

### Purpose

To evaluate the clinical applicability of intraoperative predictors for surgical outcomes after gonioscopy-assisted transluminal trabeculotomy (GATT) and microincisional trabeculectomy (MIT).

### Methods

Consecutive patients with primary, or secondary glaucoma (trauma, aphakic, or status post-retinal surgeries) with uncontrolled IOP>21mm Hg, who were scheduled to undergo GATT or MIT with or without significant cataract surgery, at a tertiary eye centre in East India between September 2021 to March 2023, were included. All surgeries were done by a single surgeon. Blanching and Trypan blue (0.4%) staining after intracameral injection using a 25 canula, were analysed in each video. The extent/pattern of blanching and blue staining in each eye was analysed objectively using an overlay of a circle with 12 sectors and a protractor tool to quantify the degrees or quadrants of blanching/staining. Multivariate regression was used to identify predictors for surgical success or the need for medications after surgery.

### Result

Of 167 eyes that were included (male: female- 134: 33), 49 eyes and 118 eyes underwent GATT and MIT, respectively, with 81 of 167 eyes undergoing concurrent cataract surgery. All eyes had a significant reduction in the number of medications after surgery. Blanching was seen in 154 of 167 eyes in a mean of 2±1.8 quadrants with 41% of eyes showing a blanching effect in >3 quadrants. Of 99 of 167 eyes where Trypan blue staining was assessed, staining in a venular, diffuse haze, or reticular pattern of staining was seen in 73 eyes, 26 eyes showed blue staining in >2 quadrants, with 16% staining in >3 quadrants. Surgical success was not predicted by the quadrants of blanching, blue staining, or other clinical variables (age, visual field, baseline intraocular pressure, type of surgery). The variables significantly predicting the need for medications included blanch (r = -0.1, p = 0.03), and blue staining (r = -0.1, p = 0.04) in <2 quadrants.

**Competing interests:** The authors have declared that no competing interests exist.

## Conclusions

Blanching and Trypan blue staining in >2 quadrants after GATT or MIT can serve as surrogate predictors for the need for medications. However more studies are mandated to find predictors for surgical success after GATT or MIT.

## Introduction

Microinvasive glaucoma surgery (MIGS) has revolutionized glaucoma surgical management over recent years [1–3]. Several conjunctival sparing surgeries, new devices, and techniques have paved a new form of minimally invasive surgeries that are safe and effective in achieving IOP control [1, 3]. While trabeculectomy still has a definite role in glaucoma, these new procedures have caused a paradigm shift in glaucoma surgery with greater adoption of these techniques in early glaucoma. Though the 2- and 5-year results suggest adequate IOP control with procedures like gonioscopy-assisted transluminal trabeculotomy (GATT), several of them require additional trabeculectomy or repeat surgeries while many require medications [1, 2, 4, 5]. Further, the long-term outcomes of visual field after these surgeries versus trabeculectomy are lacking in most studies. Surgical outcomes after MIGS and definitions of surgical success versus trabeculectomy have therefore been debated over recent years. The outflow tract is difficult to be visualized intraoperatively or preoperatively. While gonioscopy and anterior segment optical coherence tomography (AS-OCT) image the angle after MIGS, visualising the outflow tract including the collector channels, and episcleral plexus, remains a challenge [5–11]. The episcleral fluid wave or blanching effect has been proposed as a method to study the patency of the outflow tract after MIGS [12]. Some studies have reported a significant correlation between surgical success and fluorescein angiography or Trypan blue staining intraoperatively while others have used aqueous angiography or OCT-angiography for visualizing the outflow pathway after MIGS [13–17]. Yet, the sample sizes in these studies are small and the studies do not imply the clinical implication of the pattern of blue staining or extent of blanching in each eye with the clinical outcomes. This is also because of the segmental flow of aqueous in each eye and the anatomical variability in the vessels plexus draining each eye in each quadrant, this is presumably the reason why predictors for surgical success after MIGS is challenging.

We had recently introduced a new technique, Microincisional trabeculectomy (MIT), that was safe, and effective in IOP control while having less complications than reported for GATT [18]. This technique ensures a safer technique of slowly stripping the trabecular meshwork in toto without using ablation or any other tractional ripping forces like in GATT. Since this involves removing the TM in only specific quadrants as opposed to GATT where the SC is deroofed 360 degrees with a 5–0 prolene suture, the implications of this difference on the patency of the outflow pathway and the clinical outcomes is worth studying. This study evaluates and compares the pattern and extent of blanching and Trypan blue staining with the clinical outcomes after GATT and MIT.

## Methods

This was a retrospective study evaluating outcomes of GATT and MIT in adult primary, secondary, and pseudoexfoliation glaucoma (2023-164-BHR-52). This was approved by the institutional review board of LV Prasad Eye Institute, MTC campus, Bhubaneswar, and adhered to the tenets of the declaration of Helsinki with a written informed consent being taken for surgery as per institute protocols and consent for study purposes being waived for a retrospective study.

All patients diagnosed as primary open-angle or angle closure glaucoma, developmental or juvenile open-angle glaucoma, or secondary glaucoma (trauma, aphakic, or status post retinal surgeries) with uncontrolled IOP>21mm Hg or intolerant to medicines with or without significant cataract between September 2021 to March 2023, were included after a written informed consent was obtained. Patients with a history of prior conventional trabeculectomy, follow-up < 6 months, uncontrolled systemic hypertension/thyroid eye disease/orbital pathologies, or on anticoagulants, were excluded from the analysis. For those requiring concomitant cataract surgery, cataract extraction with intraocular lens insertion was done first, followed by forming the anterior chamber (AC) with ophthalmic viscoelastic devices (OVD) and then MIT/GATT. All surgeries were preferably done through the temporal corneal incision, while a superior incision was chosen only in cases of pterygium or concurrent lesion in the temporal quadrant. The choice of surgical procedure was made by the patient after counselling the risks and benefits of each procedure by the clinician.

## Surgical procedure

The surgical procedure for MIT is described elsewhere [18]. Briefly, after a goniotomy using microvitreoretinal (MVR) blade from the temporal corneal incision after intraocular lens (IOL) insertion. Now, a straight vitreoretinal scissor is directed perpendicularly to give two radial cuts at the upper trabecular edge along 4–6 clock hours in the nasal quadrant. The TM cut edge is grasped and gentle traction is applied using a 25-gauge vitreoretinal end-gripping forceps to strip it away in a single motion, **Fig 1**.

The AC is now thoroughly washed to remove viscoelastic and blood, specially at the region of the MIT. While washing, care was given to see the episcleral wave or blanching seen in the quadrant of MIT, and away from the incised site, **Fig 1**. This was now followed by the intracameral injection of Trypan blue (0.4%) under air as described in detail below.

The procedure for 5–0 prolene suture GATT is described elsewhere in detail [4, 5]. Briefly, a 5–0 prolene is threaded into eh Schlemm's canal after goniotomy with an MVR bade after which it is advanced 360 degrees of the Schlemm's canal using microforceps. The leading edge is grasped when it reaches the other cut end of the goniotomy, and the two ends are pulled in opposite directions to unroof the SC, **Fig 1**. This is followed by an AC wash and intracameral pilocarpine injection.

## Intraoperative outflow assessment

For episcleral fluid wave or blanching effect, a 25-gauge cannula from the temporal quadrant was used to inject balanced salt solution (BSS) into the nasal quadrant. A blanching effect was evidenced when the vessels filled with blood in any region disappeared with the loss of the blood column within the vessels, **Fig 1**. This would be associated with an obvious whitening or pale appearance of the sclera in that region. The quickness of the blanching response was evidenced if the blanching occurred immediately (after injecting the first 1cc) versus a forceful response where the effect was evidenced in a delayed fashion after >3cc of BSS was injected. After June 2022, this blanching response was now followed by an injection of Trypan blue injection in a 2cc syringe under air (to protect the endothelium) to visualize the effect of blue staining in vessels in any region, **Fig 1**.

## Video analysis

In each video, the eye was divided into 12 quadrants for analysis of the extent of blanch or blue staining, **Fig 2**. The quadrants of blanching and Trypan blue staining were first converted into a still image capturing a snapshot using the protractor tool available online

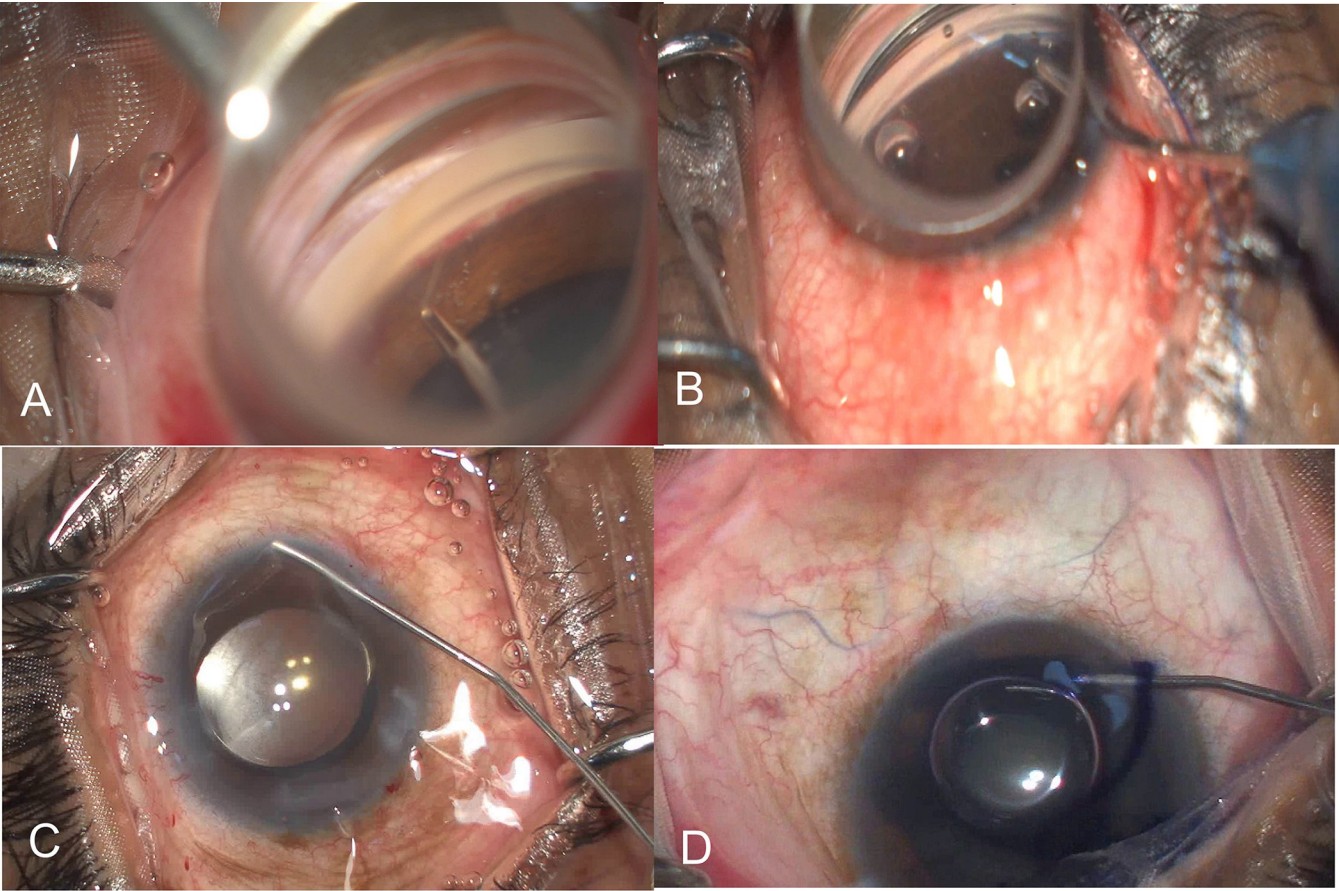

**Fig 1.** A shows the trabecular meshwork being stripped with Microforceps in microincisional trabeculectomy. B-Shows 5–0 prolene suture used for threading the Schlemm's canal and ripping the trabecular meshwork in gonioscopy-assisted transluminal trabeculotomy -See text for full description. C-shows the technique for blanching effect which is evident by loss of blood column within vessels in specific quadrants after injection of saline. D-shows the technique of Trypan blue staining after injecting the dye intracamerally-note the superficial episcleral veins that have stained with the dye (blue arrows).

(https://www.ginifab.com/feeds/angle_measurement/). The degrees of blanching and Trypan blue staining were analyzed in degrees using the protractor tool after fitting it to the eye (limbus-limbus) in the still image, as shown in **Fig 2**. The type of Trypan blue staining was classified as venular when the dye was restricted within veins located either superficially or deep (based on the location of veins either superficial or deep), as blush diffuse haze when there was no distinct pattern of stain with a bluish tinge on the sclera in that quadrant, or reticular when the staining was seen among a network or plexus of vessels in any region, **Figs 2–4**. The pattern of staining, the total number of veins and those that stained superficially or deeply), and the pattern of blue staining were noted apart from analysis on the protractor tool as described above.

Surgical success was defined as an IOP<21mm Hg without (complete success) or with (qualified success) glaucoma topical medications at 3 months postoperatively. Transient IOP spikes were defined as transient IOP >21mm Hg 1 day-3 months after surgery that resolved spontaneously or required 3–5 days of systemic acetazolamide. Topical medications were started in case of persistent IOP>21mm Hg or for achieving the clinical target range of IOP despite >30% IOP reduction after surgery. Failure was defined as IOP>21mm Hg despite maximum tolerable topical glaucoma medications at any follow-up, need for systemic

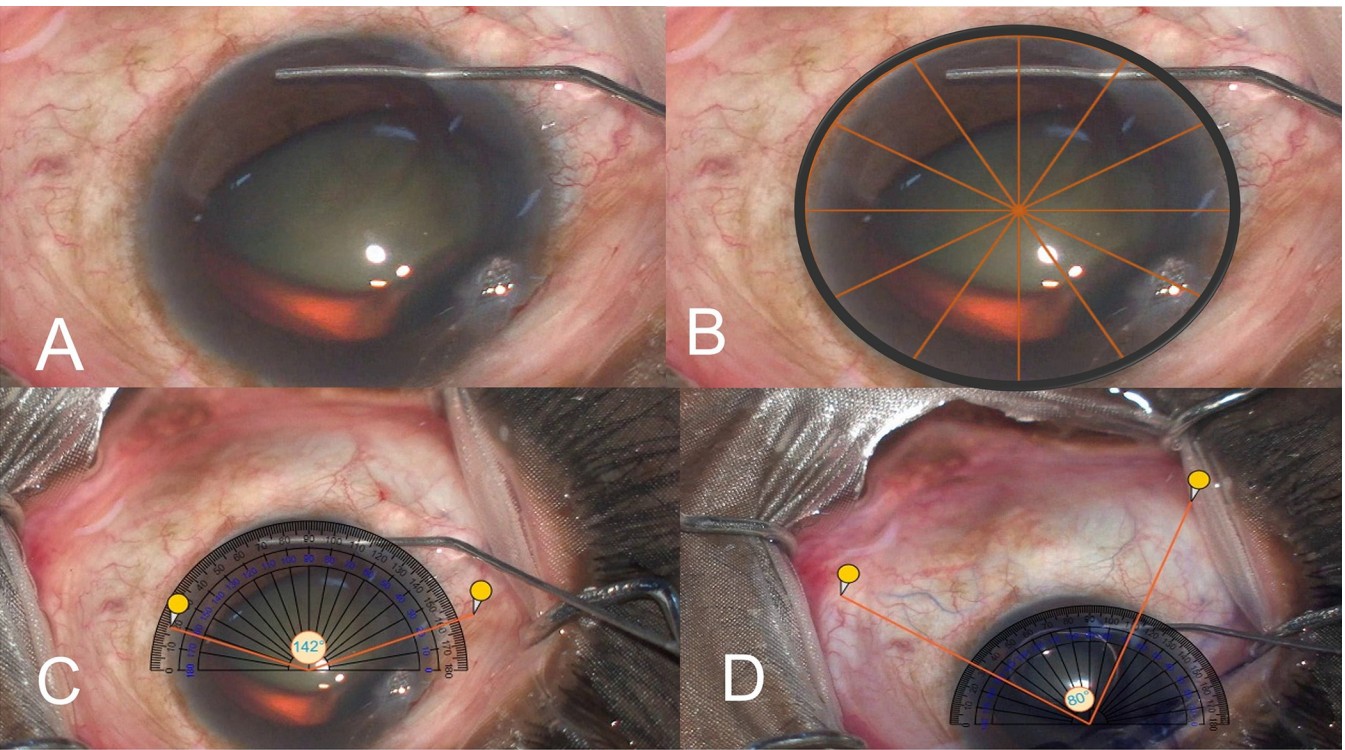

**Fig 2.** A- Shows the snapshot of the eye during surgery at the start of injection of saline for assessment of blanching. B-Shows how a circle with 12 sectors was overlaid onto the eye for calculation of extent of quadrants of blanching/blue staining in each eye. C and D- shows how the protractor tool was overlaid over the still image for calculation of degrees of blanch (C)/blue staining (D).

medications for IOP control for persistent IOP spike after 3months, the need for additional surgical procedures for IOP control, or loss of >2 lines of Snellen visual acuity.

The postoperative regimen included topical steroids in tapering doses and antibiotics for a week. Topical anti-glaucoma medications were started if the IOP was not as per the target IOP. Additional surgeries like incisional trabeculectomy were planned in the event of uncontrolled IOP despite medications.

### Statistics

The number of medications pre- and postoperatively, clinical/demographic details, highest IOP before surgery, intraoperative complications, follow-up IOP at visits, gonioscopic details at follow-up, and the need for medications/additional surgeries, ASOCT/videographic/gonioscopic correlates like peripheral anterior synechiae (PAS), adhesions, hyperreflective membranes or closure of the cleft were now analysed for the postoperative IOP outcome and surgical success. All analysis were done using Stats Corp (USA, version 13). Descriptive statistics were presented as standard deviation with mean deviation or median with interquartile range, while continuous variables were presented in proportions. Analysis between pre- and postoperative outcomes was done using paired Student's t-test or Wilcoxon sign rank test with statistical significance set at p<0.05. Clinical predictors of surgical success or failure were assessed using multivariate regression with clinical variables (age, sex, diagnosis, MD, PSD, baseline IOP, final IOP, the need for medications/surgery, extent of blanching or blue staining, type of surgery) with p<0.02 on univariate analysis considered inclusion into the multivariate model.

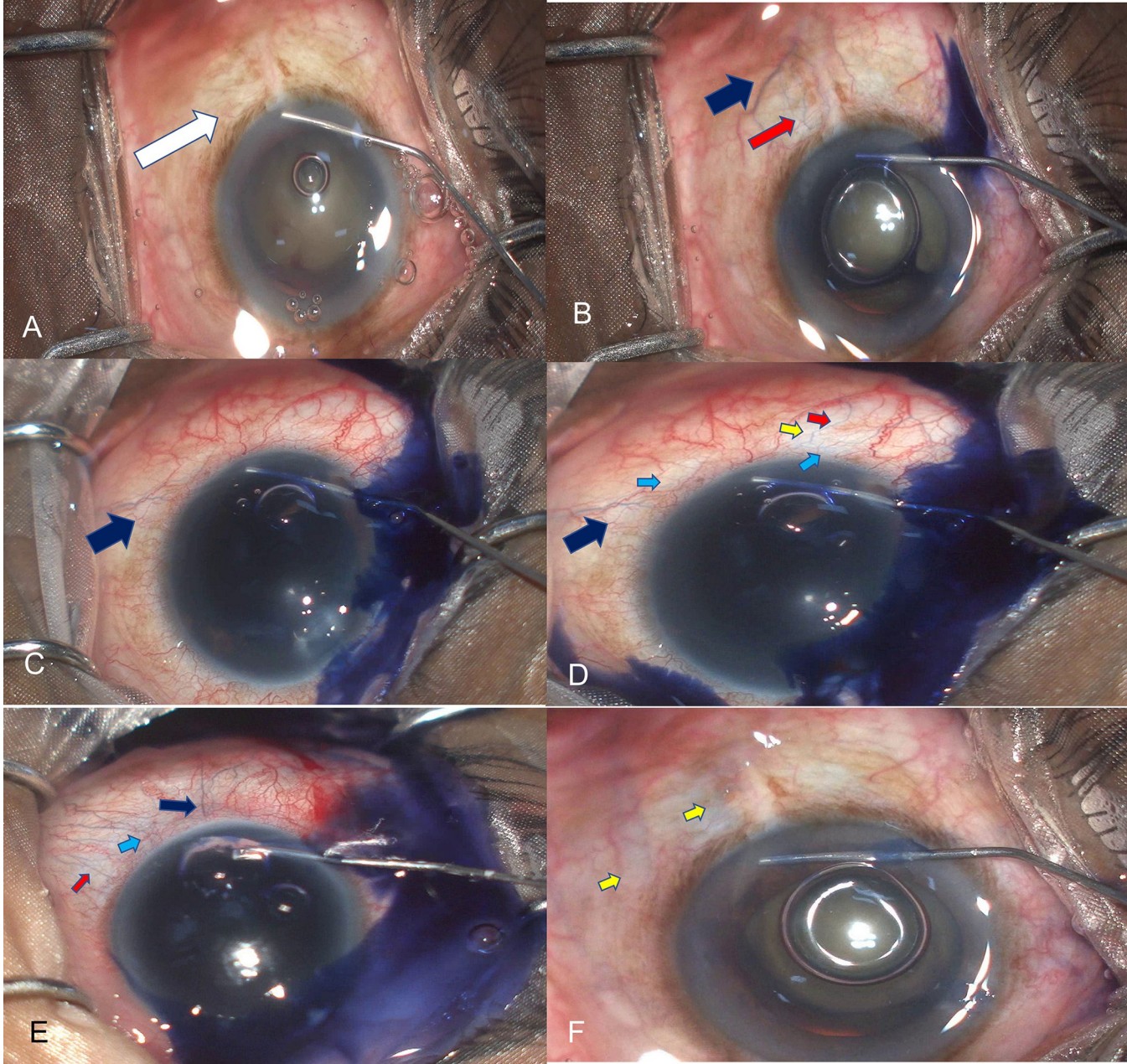

**Fig 3.** A- shows thew blanching seen in the area shown by the white arrow. B-shows the Trypan blue venular staining superficial (blue arrows) and deep episcleral (red) vessels. C and D show the progressive staining of one superficial episcleral vein that changes to a reticular pattern of the small vessel network (light blue arrow) that is associated in the inferonasal area with a bluish haze (yellow arrow). E shows venular (blue arrow of superficial vessel and red arrow of deep vessels) and reticular staining (light blue arrow). F-shows a bluish haze (yellow area) without defined staining of any superficial or dep vessel.

## Results

The number of eyes included 167 eyes (male: female- 134: 33, 85:82 right: left eye, blanch only 69 eyes operated before June 2022, blanch and Trypan blue used in 99 eyes). This included 49 eyes and 118 eyes that underwent GATT and MIT, respectively with 81 of 167 eyes undergoing concurrent cataract surgery with either procedure.

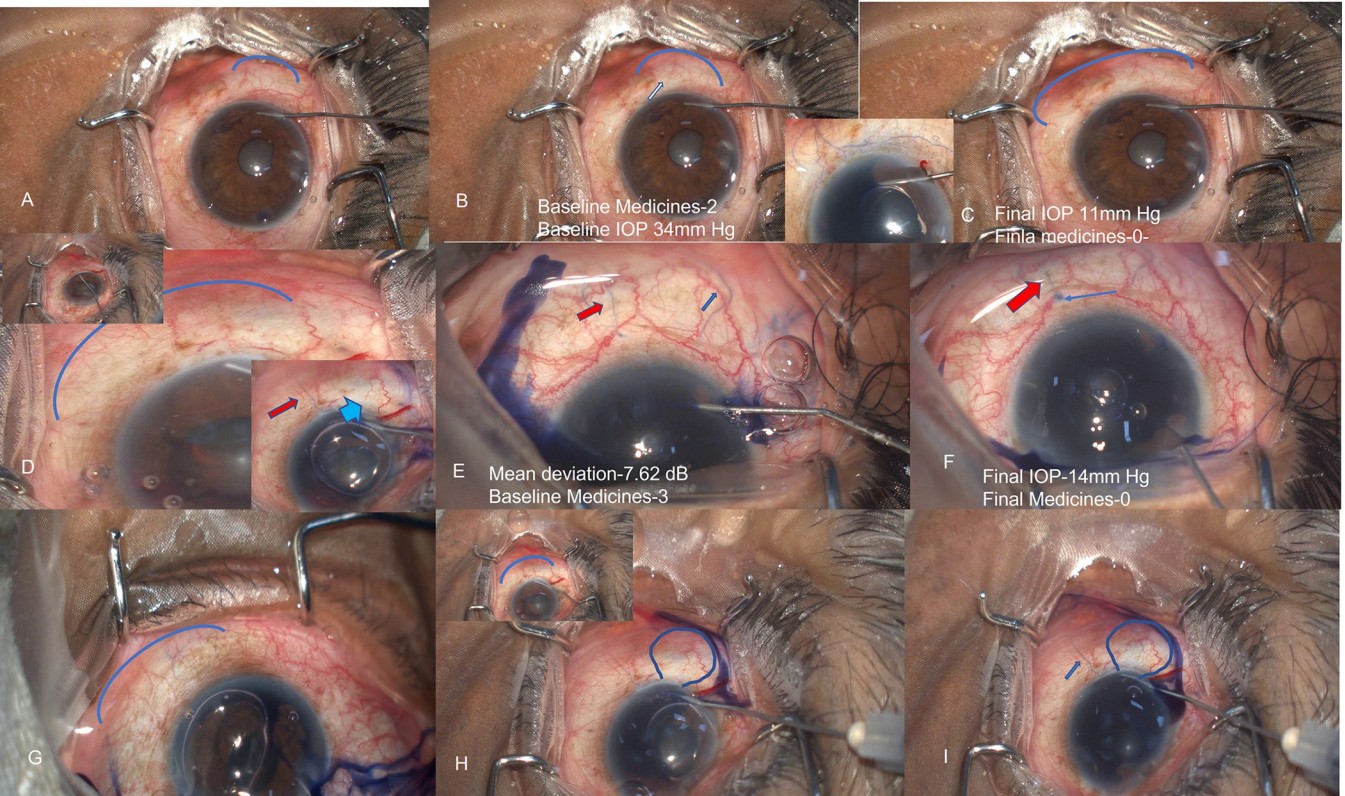

**Fig 4.** A-C shows the immediate blanching effect in 2 quadrants (A) that extends with forceful injection to 4 (B) and finally 6 quadrants in C (blue arc in each panel) with inset showing venular staining of a horizontal superficial vein. D shows the blanching (blue arc, upper inset shows the eye before the blanching) seen in 5 quadrants with the lower inset showing staining of a single deep vessel (red arrow) and adjacent bluish haze (light blue arrow). E shows the immediate venular staining of the superficial and deep vessels (blue and red arrow respectively) that further leads to staining of the perforator as the deep vein enters the sclera (blue arrow in F). G-I show the blanching seen in 3 quadrants (blue arc) compared to the blush haze seen in a different region (marked balloon in H) and finally a delayed venular staining of a superficial vessel (blue arrow in I).

The baseline parameters including baseline IOP, IOP at the time of surgery, and the number of medicines were not significantly different between the eyes that underwent GATT versus MIT, **S1 Table**.

IOP spikes were seen in 28 eyes (16 GATT and 12 MIT), while macrohyphema was seen in 6 eyes of GATT and 2 of MIT eyes. All IOP spikes within <1 month after surgery resolved with transient systemic acetazolamide tablets for 1 week or topical medications, with no eye requiring AC wash. The final IOP was not significantly different between eyes that underwent GATT versus MIT. All eyes experienced a significant reduction in the IOP at 1day and at the final follow-up (14±2.1months, r = 12–18), with a significant reduction in the number of medications, **Table 1**.

## Outflow assessment

**Table 2** gives the details of quadrants of blanching and patterns of blue staining seen in these eyes. Blanching was seen in 154 of 167 eyes in a mean of 2±1.8 quadrants with 41% of eyes showing a blanching effect in >3 quadrants with 123 eyes showing an immediate blanching effect. There were differences in the onset of different types of patterns in different eyes, with some patterns emerging with delayed onset in quadrants similar or different to that of the quadrants where blanching was seen, **Fig 4**. One eye that underwent MIT in 6 clock hours

**Table 1. Clinical demographic profile of patients that underwent GATT and MIT in the study.** See S1 Table for comparisons between GATT and MIT.

| Variables | Mean+SD or N | |
|---|---|---|
| Age (years) | 58±28.4 | |
| Male:Female | 133:34 | |
| Diagnosis | JOAG | 15 |
| | POAG/PACG | 72/16 |
| | PXG | 52 |
| | Secondary glaucoma | |
| | Steroid | 2 |
| | S/p VR sx | 2 |
| | Aphakic glaucoma | 1 |
| | Uveitic glaucoma | 1 |
| | Angle recession | 5 |
| BIOP | 21±9.2 | |
| Baseline MD | -17±10.1 | |
| Baseline PSD | 7±3.5 | |
| Baseline number of medicines | 2±1.2 | |
| Final meds | 0.3±0.7 | |

JOAG-Juvenile open-angle glaucoma, POAG-primary open-angle glaucoma, PACG-primary angle closure glaucoma; PXG-pseudoexfoliation glaucoma; MD-Mean deviation, PSD- pattern standard deviation; IOP-intraocular pressure; VR sx-Vitreoretina surgery

showed immediate blanching in 3 quadrants (12–3 clock) that extended over the 3–6 clock hour with forceful injection over a time, **Fig 4**. The quadrants of blanching were not significantly different between eyes that underwent GATT or MIT (3±1.8 and 3± 1.09, respectively), S2 Table.

Correlating the quadrant of blanching to the final IOP or need for medications, the quadrant of blanching<2 quadrants correlated well with the need for medications after GATT or MIT. **Fig 4** and **S1 Fig** show some representative cases with the IOP at the final follow-up.

Absence of blanching was seen in 13 eyes which included 3 angle recession glaucoma, 5 advanced JOAG, 2 POAG and 1 PACG eye. Of these, 4 eyes required 1–3 medications while 1 angle recession glaucoma required trabeculectomy for pressure control.

## Trypan blue staining

Trypan blue staining after the blanching effect was assessed in 99 eyes. Of 99 eyes, 26 eyes did not show any blue staining with 7 eyes not showing blanching also. Of 73 eyes, 26 eyes showed blue staining in >2 quadrants with 16% staining in >3 quadrants, **Table 2**. The pattern of staining was venular or predominantly venular in 43 eyes while in 17 eyes, the venular pattern also had associated reticular and haze patterns of staining. Four eyes had only a bluish diffuse haze pattern of blue staining without a definite pattern or venular filling in any quadrant. The mean number of veins in 73 eyes that were stained by a single injection was 2±2.5 with 9 eyes having >3 veins per eye staining with a single injection. The number of superficial veins staining in each quadrant ranged from 2–3 superficial veins and 2–4 deep veins with 4 eyes showing>3 deep vein/eye staining. Of 26 eyes that did not show blue staining, 3 eyes required 2 medicines, and 1 required 3 medicines for IOP control.

The quadrants of blanching correlated well with quadrants of venular staining, p = 0.01, r = 0.3. while it did not correlate with deep, or haze pattern of staining.

**Table 2. Extent of blanching and Trypan blue staining observed after GATT or MIT in the study.** See S1 Table for comparisons between GATT and MIT.

| Variable studied | Quadrants of observed variable (among 12 quadrants)#- No of eyes | | | Degrees as measured# Median (range) |
|---|---|---|---|---|
| | **or** | | | |
| | number of quadrants | - | number of eyes | |
| Blanching (N = 167) | 0 | - | 13 | 47.2(0–188) |
| | 1 | - | 8 | |
| | 2 | - | 69 | |
| | 3 | - | 36 | |
| | 4 | - | 17 | |
| | 5 | - | 12 | |
| | 6 | - | 11 | |
| | 8 | - | 1 | |
| Trypan blue staining (N = 99) | 0 | - | 26 | 32.8 (0–65.6) |
| | 1 | - | 13 | |
| | 2 | - | 34 | |
| | 3 | - | 10 | |
| | 4 | - | 6 | |
| | 5 | - | 10 | |
| Pattern of Trypan blue staining* | Venular | 43 | | NA |
| | Venular+reticular | 2 | | |
| | Venular+haze | 5 | | |
| | Haze | 4 | | |
| | Venular+reticular+haze | 15 | | |
| | Reticular+haze | 4 | | |

#-See text for the method for measuring the extent of blanch or blue staining

*-See text for a detailed description of each pattern of blue staining.

Surgical success was seen in 153 eyes (complete success in 130 eyes, 14 eyes requiring 2 medicines, 8 eyes requiring 1 medicine, 1 eye requiring 3 medicines), and failure was seen in 13 eyes. Of 13 eyes that failed, 6 eyes required additional trabeculectomy for controlling pressure.

While surgical success (complete or qualified) was not predicted by the quadrants of blanching, blue staining, or other clinical variables (age, MD PSD, baseline IOP, type of surgery), the variables significantly predicting the need for medications on multivariate regression included quadrants of blanch (r = -0.1, p = 0.03), and the quadrants of blue staining (r = -0.1, p = 0.04) in <2 quadrants, S3 Table. The diagnosis, number of quadrants of MIT/GATT, or the severity of the damage did not influence the need for medications after either surgery, while the number of superficial venular staining in >2 quadrants showed a trend toward significant association with the need for additional medicines (p = -0.1, p = 0.055). Other patterns of staining were not predictive of surgical success or failure after GATT or MIT.

## Discussion

Trypan blue staining and blanching have been known as modes for testing the patency of the outflow apparatus after MIGS [6, 7, 9, 11, 14]. This study found blanching and Trypan blue staining seen in <2 quadrants (and to a lesser extent the extent of superficial venular staining<2 quadrants) intraoperatively to be predictive of the need for medicines after surgery.

Angle recession, advanced JOAG, and developmental glaucoma were the most common causes for the need for additional surgery and the absence of intraoperative blanching or blue staining in this study. The quadrants of MIT done, the pattern of Trypan blue staining, the severity of glaucoma damage at the time of surgery, or the baseline IOP, did not predict the surgical success or the need for medications.

The means of evaluating the outflow system include aqueous angiography, fundus fluorescein angiography, aqueous angiography, 3-D visualization, and optical coherence tomography angiography [8–17, 19–25]. Yet, these either require specific filters on the operative microscope or require portable machines with additional features for intraoperative use and therefore are not applicable universally. Trypan blue offers a cost-effective and viable option for assessing the outflow under direct visualization after the surgery under the operating microscope [14]. Trypan blue staining using an endoscopic catheter into the Schlemm's canal has been reported to predict the patency of the outflow apparatus after goniotomy or canaloplasty. One study has looked at different patterns of blue staining seen in 4 eyes after MIGS [24]. Yet, the implications and the clinical significance of these patterns of blanching or staining have not been evaluated earlier. This study revealed that staining <2 quadrants predicts the future need for glaucoma medications, suggesting, that the anatomical variations out the venous plexus draining each quadrant in each eye may be one of the causes of failure after MIGS procedures or filtering surgeries. Non-invasive modalities for studying the normal anatomical variations, the pattern of segmental aqueous flow in each quadrant before surgery need to be explored to study the surgical and clinical outcomes after MIGS.

Assessment of the outflow system after MIGS is crucial to understand the causes of failure or success after any MIGS procedure [7, 9, 11, 21, 26, 27]. Blanching represents the episcleral fluid wave that represents a patent outflow pathway. An immediate blanching obviously represents full patency of the outflow system, while a delayed blanching after forceful injection, represents a partial collapse of the collector channels or the veins draining each segment. The absence of blanching indicates a closed outflow system which may be a surrogate for failure after any MIGS procedure. This study found that the absence of blanching in <2 quadrants may be used as an intraoperative surrogate for the need for additional medications after surgery. This is valuable for eyes with severe damage at presentation or secondary glaucoma, where the outflow system in many quadrants would be damaged. A closer follow-up of IOP postoperatively may be advisable in these eyes.

Superficial versus deep vessels stained in this procedure show a lot of variabilities, though the predominant staining pattern was superficial venular staining. In normal eyes, the anterior episcleral vessel may course superficially for some distance before entering the sclera (deep episcleral vessels) [25, 28]. The deep vessels anastomose with the networks/twigs from adjacent anterior ciliary arteries. The episcleral plexus arises for branches from the superficial or deep branches of the anterior episcleral circle with some isolated twigs from the sclera that rise close to the limbus. Studies using angiography have shown that the deeper branches stain faintly or show a haze on angiography as seen in our study in some cases [21, 24–26]. Yet, the different patterns of staining and the different number of quadrants of blanching or staining seen in this study suggest anatomical variability in the outflow system as one of the causes for the variable results after MIGS. Future studies using non-invasive in-vivo analysis of pre-and postoperative changes in the vessel network may give insights into how the outflow system responds to any surgical procedures. This may also identify novel predictors in the outflow system that cause failure after MIGS.

This study had several limitations. This was part of a prospective study evaluating the blanching/staining patterns after GATT and MIT. GATT involves creating a resistance-free path in 360 degrees while MIT involves the same in specific quadrants which potentially

should be confounding the surgical success. Yet, we did not find this to be significantly different results of blanching or blue staining in GATT or MIT eyes. We excluded those eyes with uncontrolled hypertension or systemic disease that causes severe bleeding during any MIGS-obviously, the episcleral venous pressure (EVP) may be higher in these cases suggesting possible differences in the functioning of the outflow system and therefore the pattern of blanching or staining in these eyes. We did not include the quantitative evaluation of the angle using anterior segment imaging. Yet, we do not believe that this would have impacted the results largely. OCT angiography, the most robust measure for the outflow system, was not used in this study. Nevertheless, our results give interesting insights into the clinical implications of blanching/staining patterns that can be used as an intraoperative indirect surrogate fr predicting the postoperative need for medications after GATT or MIT.

## Conclusion

Blanching and Trypan blue intracameral injection can serve as potential intraoperative predictors for the need for glaucoma medications after minimally invasive glaucoma surgery like MIT and GATT. Anatomical variations in the pattern of veins draining each quadrant in each eye make it challenging to predict surgical success after MIGS. In the absence of anterior segment imaging to assess the outflow pathway, blanching and Trypan blue can be surrogates for optimizing surgical outcomes in MIGS for any clinician.

## Supporting information

**S1 Fig.** A-C shows the progressive blanching seen in 2 quadrants (arc) to 6 quadrants in C. The staining pattern in this eye was restricted to deep venular (red arrow) and blush haze (light blue arrow) staining in small regions in different quadrants.
(JPG)

**S1 Table. Comparison of the clinical profile of patients that underwent GATT or MIT.** See text for full description.
(DOCX)

**S2 Table. Comparisons of blanching and blue staining for intraoperative outflow assessment after GATT or MIT.** (See text for full description).
(DOCX)

**S3 Table. Multivariate regression of clinical variables predicting the need for medicines after GATT or MIT.** See text for full description.
(DOCX)

## Acknowledgments

Hyderabad Eye Research Foundation.

## Author Contributions

**Conceptualization:** Aparna Rao.

**Data curation:** Aparna Rao, Sujoy Mukherjee.

**Formal analysis:** Aparna Rao.

**Investigation:** Aparna Rao, Sujoy Mukherjee.

**Methodology:** Aparna Rao.

**Project administration:** Aparna Rao.

**Resources:** Aparna Rao, Sujoy Mukherjee.

**Software:** Aparna Rao.

**Supervision:** Aparna Rao.

**Validation:** Aparna Rao.

**Visualization:** Aparna Rao, Sujoy Mukherjee.

**Writing – original draft:** Aparna Rao.

**Writing – review & editing:** Aparna Rao, Sujoy Mukherjee.

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
