## [Decision Letter · Decision Letter 0]

15 Aug 2023

PONE-D-23-18067‘Intraoperative predictors for clinical outcomes after microinvasive glaucoma surgery”PLOS ONE

Dear Dr. Rao,

Thank you for submitting your manuscript to PLOS ONE. After careful consideration, we feel that it has merit but does not fully meet PLOS ONE’s publication criteria as it currently stands. Therefore, we invite you to submit a revised version of the manuscript that addresses the points raised during the review process.

ACADEMIC EDITOR: The manusctipt is well drafted, however there is scope of improvement.

We look forward to receiving your revised manuscript.

Kind regards,

Natasha Gautam, MBBS, MS

Academic Editor

PLOS ONE

3. We notice that your supplementary figure is uploaded with the file type 'Figure'. Please amend the file type to 'Supporting Information'. Please ensure that each Supporting Information file has a legend listed in the manuscript after the references list.

Additional Editor Comments:

It is a well written manuscript describing the predictors for intraoperative success after MIGS.

The criteria for qualified success/ failure is not clear. The authors have mentioned maximum tolerable glaucoma medications. Did they count only topical medications or also counted oral medications in defining qualified success?

In the results section, last paragraph, the authors mentioned that intraoperative stain in > 2 quadrants was associated with higher number of medications. It should be vice a versa.

The authors have commented on final follow up while decribing results. They should mention if they are mentioning results at 1 year follow up, or should define what was the mean and median follow up if they want to write the final follow up.

Reviewers' comments:

Reviewer's Responses to Questions

**Comments to the Author**

1. Is the manuscript technically sound, and do the data support the conclusions?

Reviewer #1: Yes

Reviewer #2: Partly

2. Has the statistical analysis been performed appropriately and rigorously? 

Reviewer #1: Yes

Reviewer #2: No

3. Have the authors made all data underlying the findings in their manuscript fully available?

Reviewer #1: Yes

Reviewer #2: Yes

4. Is the manuscript presented in an intelligible fashion and written in standard English?

Reviewer #1: Yes

Reviewer #2: Yes

5. Review Comments to the Author

Reviewer #1: Thank you for submitting your article. Overall, I found the paper to be well-written and informative. However, I have a few suggestions that I believe would improve the quality of the article. Please address these minor corrections before resubmitting your work. Overall, I believe these minor corrections will enhance the clarity and impact of your article.

Reviewer #2: The author launch an interesting study about MIT and GATT, which want to find the predictor factor of these MIGS proceduere. The details of the study, statistical methods, still need to be clarified before the article is published:

1) According to the author's narratives in limitation paragraphs, GATT is 360 degree trabcular meshwork cutting while MIT is a quadrant incision method. The blanching or staining difference between these two group should be analized in the manuscript. And the result of surgical success ( complete or qualified ) should also comparing between the two groups.

2) The author use multivariiate regression module to analysis the predict facotr of surgical success.But the details table of this statistical analysis is not presented in the manuscript.

3) Whether the set position of the protractor is fixed, or based on the TM incision position (especially in MIT surgery), or the position where the irrigation needle is placed ? If it is only a partial incision (MIT), then the quadrants of blanching or staining will actually be affected by the incision site.

6. PLOS authors have the option to publish the peer review history of their article (what does this mean?). If published, this will include your full peer review and any attached files.

Reviewer #1: **Yes: **Mahmood Ali

Reviewer #2: No

---

## [Author Response · Author response to Decision Letter 0]

30 Aug 2023

¬¬To,

The Editor,

Dear Sir/Madam,

We are pleased to submit our revised manuscript titled "Intraoperative Predictors for Clinical Outcomes after Microinvasive Glaucoma Surgery." Additionally, we have had the manuscript professionally edited. We have taken care to address each of the reviewers' comments with detailed point-to-point clarifications. We remain open to any further suggestions or feedback that could contribute to enhancing the quality of the manuscript. Thank you for your consideration.

All the authors have contributed equally towards the preparation of the manuscript and have no financial or proprietary interest in the products used in the study. We also declare that this article has not been published previously or is under review with any other journal.

a) All acknowledgments and financial disclosures/funding information is included in the manuscript. We would like to clarify that this acknowledgment is for the research foundation supporting all the institute's healthcare-related studies. This does not entail funding from the foundation. This study did not receive any funding from any agency or organization. 

b) All data have been given in the manuscript with additional patient-identifying information that may be shared after consent upon request. We have also made available a minimal dataset along with all relevant data that has already been shared in the supplemental data. 

Thanking you

Answer; we do confirm the same

Answer: This has already been provided, as per instructions.

3. We notice that your supplementary figure is uploaded with the file type 'Figure'. Please amend the file type to 'Supporting Information'. Please ensure that each Supporting Information file has a legend listed in the manuscript after the references list.

Answer: we have changed this now to supporting information, as suggested.

Additional Editor Comments:

It is a well written manuscript describing the predictors for intraoperative success after MIGS.

The criteria for qualified success/ failure is not clear. The authors have mentioned maximum tolerable glaucoma medications. Did they count only topical medications or also counted oral medications in defining qualified success?

In the results section, last paragraph, the authors mentioned that intraoperative stain in > 2 quadrants was associated with higher number of medications. It should be vice a versa.

The authors have commented on final follow up while describing results. They should mention if they are mentioning results at 1 year follow up, or should define what was the mean and median follow up if they want to write the final follow up.

Answer: We deeply appreciate the insightful points of the editor that we have now addressed in the revised manuscript. Additionally, we have clarified the criteria for success to eliminate any potential confusion for readers. Moreover, we have incorporated the suggested information regarding the final follow-up duration. Thank you for your valuable guidance.

Reviewers' comments:

5. Review Comments to the Author

Reviewer #1: Thank you for submitting your article. Overall, I found the paper to be well-written and informative. However, I have a few suggestions that I believe would improve the quality of the article. Please address these minor corrections before resubmitting your work. Overall, I believe these minor corrections will enhance the clarity and impact of your article.

Answer: We express our gratitude to the reviewer for their encouragement and sincerely appreciate their valuable suggestions. We welcome any further recommendations to enhance the manuscript and are truly thankful for the significant improvements made by incorporating these insightful suggestions.

Reviewer #2: The author launch an interesting study about MIT and GATT, which want to find the predictor factor of these MIGS proceduere. The details of the study, statistical methods, still need to be clarified before the article is published:

1) According to the author's narratives in limitation paragraphs, GATT is 360 degree trabcular meshwork cutting while MIT is a quadrant incision method. The blanching or staining difference between these two group should be analized in the manuscript. And the result of surgical success ( complete or qualified ) should also comparing between the two groups.

Answer: We certainly concur that a comparison of the blanching or staining method between the two approaches would be beneficial. Furthermore, it's important to highlight that we discovered no discernible difference in blanching patterns between GATT and MIT (the same reflected in the multivariate analysis with the type of surgery not defining the blanching or blue staining pattern). However, we wish to clarify that our intention was not to conduct a direct comparative study between GATT and MIT. Instead, our focus was on developing a simplified method for assessing the outflow pathway. A comparative analysis between these two surgeries, considering the variation in the number of quadrants operated, could introduce confounding factors. For instance, a greater extent of deroofing is logically expected to lead to a more substantial reduction in outflow resistance though this was not seen in this study. This consideration influenced our decision not to pursue a comparative study in this research. This information has already been detailed in the results section (outflow assessment subheading), indicating that a comparative analysis of this nature would yield similar outcomes. Yes, we are comparing the prospective clinical outcomes between GATT and MIT-we would be delighted to incorporate this suggestion to see the differences in blanching/blue staining patterns with other anterior segment imaging parameters in that study. 

2) The author use multivariiate regression module to analysis the predict facotr of surgical success. But the details table of this statistical analysis is not presented in the manuscript.

Answer: We wish to provide clarity regarding the inclusion of multivariate regression details and considered variables. These specifics have been outlined in the manuscript's Results section, specifically in the final paragraph. We are more than willing to share any additional information pertaining to this matter. Given that a significant association was not observed for most variables, we made the decision not to present this information in a table or figure and rather included the details in the text.

3) Whether the set position of the protractor is fixed, or based on the TM incision position (especially in MIT surgery), or the position where the irrigation needle is placed ? If it is only a partial incision (MIT), then the quadrants of blanching or staining will actually be affected by the incision site.

Answer: We want to provide further clarification regarding the protractor tool mentioned in our study. It's important to note that the protractor tool we utilized is a virtual online tool that is superimposed onto the image or video. It doesn't involve physically placing the tool on the eye during surgery. This tool is positioned over the eye in a manner that spans from limbus to limbus. Consequently, its placement is not influenced by factors such as the incision site or any similar considerations.

---

## [Decision Letter · Decision Letter 1]

2 Oct 2023

PONE-D-23-18067R1‘Intraoperative predictors for clinical outcomes after microinvasive glaucoma surgery”PLOS ONE

Dear Dr. Rao,

Thank you for submitting your manuscript to PLOS ONE. After careful consideration, we feel that it has merit but does not fully meet PLOS ONE’s publication criteria as it currently stands. Therefore, we invite you to submit a revised version of the manuscript that addresses the points raised during the review process. Please submit your revised manuscript by Nov 16 2023 11:59PM. If you will need more time than this to complete your revisions, please reply to this message or contact the journal office at plosone@plos.org. Please include the following items when submitting your revised manuscript:A rebuttal letter that responds to each point raised by the academic editor and reviewer(s). You should upload this letter as a separate file labeled 'Response to Reviewers'.A marked-up copy of your manuscript that highlights changes made to the original version. You should upload this as a separate file labeled 'Revised Manuscript with Track Changes'.An unmarked version of your revised paper without tracked changes. You should upload this as a separate file labeled 'Manuscript'.

We look forward to receiving your revised manuscript.

Kind regards,

Natasha Gautam, MBBS, MS

Academic Editor

PLOS ONE

Journal Requirements:

Additional Editor Comments:

The authors have done a great job in addressing the comments. However they are advised to describe the results of GATT and MIT separately in Table 1 and 2, even if they want to avoid comparative analysis. Because the eyes with trypan blue stain and blanching would be different in eyes with MIT and GATT, due to difference in quadrants with trabecular meshwork excusiom. Therefore it would be nice to present the results of two groups differently. Its the author's choice if they don't want to compute statstical significant results between two groups or not, since that is not the primary intention of the study.

Reviewers' comments:

Reviewer's Responses to Questions

**Comments to the Author**

1. If the authors have adequately addressed your comments raised in a previous round of review and you feel that this manuscript is now acceptable for publication, you may indicate that here to bypass the “Comments to the Author” section, enter your conflict of interest statement in the “Confidential to Editor” section, and submit your "Accept" recommendation.

Reviewer #1: All comments have been addressed

Reviewer #2: (No Response)

2. Is the manuscript technically sound, and do the data support the conclusions?

Reviewer #1: Yes

Reviewer #2: Partly

3. Has the statistical analysis been performed appropriately and rigorously? 

Reviewer #1: Yes

Reviewer #2: No

4. Have the authors made all data underlying the findings in their manuscript fully available?

Reviewer #1: Yes

Reviewer #2: Yes

5. Is the manuscript presented in an intelligible fashion and written in standard English?

Reviewer #1: Yes

Reviewer #2: Yes

6. Review Comments to the Author

Reviewer #1: The article is well-organized and presents a valuable contribution to the field of microinvasive glaucoma surgery. The methodology is robust, but potential limitations, such as the retrospective nature of the study and the sample size, are appropriately acknowledged. The discussion effectively connects the findings to broader clinical implications.

Note:

A few grammatical errors in the article have been identified.

Original Text: "All eyes had a significant reduction in the number of medications after surgery."

Suggested Correction: "All eyes experienced a significant reduction in the number of medications after surgery."

Original Text: "The means of evaluating the outflow system include aqueous angiography, fundus fluorescein angiography, aqueous angiography, 3-D visualization, and optical coherence tomography angiography."

Suggested Correction: "The means of evaluating the outflow system include aqueous angiography, fundus fluorescein angiography, 3-D visualization, and optical coherence tomography angiography."

Original Text: "Trypan blue is a cheap and effective alternative means of assessing the outflow under direct visualization after the surgery under the operating microscope."

Suggested Correction: "Trypan blue offers a cost-efficient and viable option for assessing outflow under direct visualization following surgery with the aid of the operating microscope."

Reviewer #2: 1）Since the author has listed two different surgical methods, the results should still be presented according to different surgical types, rather than mixed together. Regardless of whether there is a statistical difference between the two groups, the relevant data in Table 1 and 2 needs to be clearly presented to the reader.

2）The results of the multi variate regression analysis also need to be presented, even in the appendix section.

3）Why choose p<0.02 as the judgment criterion in the statistical method section. In general, this standard is either p<0.01 or p<0.05.

4）Regarding this question of "3) Why the set position of the predictor is fixed, or based on the TM decision position (specifically in MIT summer)...", I would like to know whether the author judges it based on fixed quadrants or hours site? If the range of blank is 5-7 hours, it is 3 hours (equivalent to a quadrant) calculated by hour position; If calculated based on fixed quadrants, it is 2 quadrants.

7. PLOS authors have the option to publish the peer review history of their article (what does this mean?). If published, this will include your full peer review and any attached files.

Reviewer #1: **Yes: **Mr. Mahmood Ali

Reviewer #2: No

---

## [Author Response · Author response to Decision Letter 1]

2 Oct 2023

To,

The Editor,

Dear Sir/Madam,

We are pleased to submit our revised manuscript titled "Intraoperative Predictors for Clinical Outcomes after Microinvasive Glaucoma Surgery." Additionally, we have had the manuscript professionally edited. We have taken care to address each of the reviewers' comments with detailed point-to-point clarifications. We remain open to any further suggestions or feedback that could contribute to enhancing the quality of the manuscript. Thank you for your consideration.

All the authors have contributed equally towards the preparation of the manuscript and have no financial or proprietary interest in the products used in the study. We also declare that this article has not been published previously or is under review with any other journal.

a) All acknowledgments and financial disclosures/funding information is included in the manuscript. We would like to clarify that this acknowledgment is for the research foundation supporting all the institute's healthcare-related studies. This does not entail funding from the foundation. This study did not receive any funding from any agency or organization. 

b) All data have been given in the manuscript with additional patient-identifying information that may be shared after consent upon request. We have also made available a minimal dataset along with all relevant data that has already been shared in the supplemental data. 

Thanking you

Round 2

Associate editor:

The authors have done a great job in addressing the comments. However they are advised to describe the results of GATT and MIT separately in Table 1 and 2, even if they want to avoid comparative analysis. Because the eyes with trypan blue stain and blanching would be different in eyes with MIT and GATT, due to difference in quadrants with trabecular meshwork excusiom. Therefore it would be nice to present the results of two groups differently. Its the author's choice if they don't want to compute statstical significant results between two groups or not, since that is not the primary intention of the study.

Answer: We would like to clarify that as stated in the manuscript, trypam blue was done only after June so included only 99 eyes. We have therefore presented the blanching/staining between the GATT and MIT eyes in a supplementary table for logistic reasons of including that information in table 1 or 2 and to avoid confusion for readers when talking about outflow assessment in general. 

Review Comments to the Author

Reviewer #1: 

The article is well-organized and presents a valuable contribution to the field of microinvasive glaucoma surgery. The methodology is robust, but potential limitations, such as the retrospective nature of the study and the sample size, are appropriately acknowledged. The discussion effectively connects the findings to broader clinical implications.

Answer: We thank the reviewer for the approval and would welcome any further suggestions. 

:

A few grammatical errors in the article have been identified.

Original Text: "All eyes had a significant reduction in the number of medications after surgery."

Suggested Correction: "All eyes experienced a significant reduction in the number of medications after surgery."

Original Text: "The means of evaluating the outflow system include aqueous angiography, fundus fluorescein angiography, aqueous angiography, 3-D visualization, and optical coherence tomography angiography."

Suggested Correction: "The means of evaluating the outflow system include aqueous angiography, fundus fluorescein angiography, 3-D visualization, and optical coherence tomography angiography."

Original Text: "Trypan blue is a cheap and effective alternative means of assessing the outflow under direct visualization after the surgery under the operating microscope."

Suggested Correction: "Trypan blue offers a cost-efficient and viable option for assessing outflow under direct visualization following surgery with the aid of the operating microscope."

Answer: All the grammatical corrections have been made, as suggested by the reviewer. 

Reviewer #2: 

1）Since the author has listed two different surgical methods, the results should still be presented according to different surgical types, rather than mixed together. Regardless of whether there is a statistical difference between the two groups, the relevant data in Table 1 and 2 needs to be clearly presented to the reader.

Answer: We have included a separate table S2 on results for GATT and MIT with regards to blanching since blue staining was done only after June 2022 (n=99 eyes). 

2）The results of the multi variate regression analysis also need to be presented, even in the appendix section.

Answer: As suggested, we have now included a supplementary table S3 on the multivariate analysis.

3）Why choose p<0.02 as the judgment criterion in the statistical method section. In general, this standard is either p<0.01 or p<0.05.

Answer: While doing a multivariate analysis, it is preferable to include only those variables that have a significant association on a univariate analysis that makes the statistical comparison robust. In this procedure, we include only those that show a significant association on univariate rather than p<0.02 so that spurious association on a multivariate model is avoided which is not uncommon if we include a lot of univariate variables with weak associations (It may be common that p<0.05 may give significance ion multivariate model while variables on univariate may not be statistically significant). This is the reason we also chose p<0.02 as that mark for inclusion into the multivariate model. 

4）Regarding this question of "3) Why the set position of the predictor is fixed, or based on the TM decision position (specifically in MIT summer)...", I would like to know whether the author judges it based on fixed quadrants or hours site? If the range of blank is 5-7 hours, it is 3 hours (equivalent to a quadrant) calculated by hour position; If calculated based on fixed quadrants, it is 2 quadrants.

Answer: We agree and clarify that all the comparisons were made for quadrants and not clock hours as stated in the manuscript.

Review round 1

Answer; we do confirm the same

Answer: This has already been provided, as per instructions.

3. We notice that your supplementary figure is uploaded with the file type 'Figure'. Please amend the file type to 'Supporting Information'. Please ensure that each Supporting Information file has a legend listed in the manuscript after the references list.

Answer: we have changed this now to supporting information, as suggested.

Additional Editor Comments:

It is a well written manuscript describing the predictors for intraoperative success after MIGS.

The criteria for qualified success/ failure is not clear. The authors have mentioned maximum tolerable glaucoma medications. Did they count only topical medications or also counted oral medications in defining qualified success?

In the results section, last paragraph, the authors mentioned that intraoperative stain in > 2 quadrants was associated with higher number of medications. It should be vice a versa.

The authors have commented on final follow up while describing results. They should mention if they are mentioning results at 1 year follow up, or should define what was the mean and median follow up if they want to write the final follow up.

Answer: We deeply appreciate the insightful points of the editor that we have now addressed in the revised manuscript. Additionally, we have clarified the criteria for success to eliminate any potential confusion for readers. Moreover, we have incorporated the suggested information regarding the final follow-up duration. Thank you for your valuable guidance.

Reviewers' comments:

5. Review Comments to the Author

Reviewer #1: Thank you for submitting your article. Overall, I found the paper to be well-written and informative. However, I have a few suggestions that I believe would improve the quality of the article. Please address these minor corrections before resubmitting your work. Overall, I believe these minor corrections will enhance the clarity and impact of your article.

Answer: We express our gratitude to the reviewer for their encouragement and sincerely appreciate their valuable suggestions. We welcome any further recommendations to enhance the manuscript and are truly thankful for the significant improvements made by incorporating these insightful suggestions.

Reviewer #2: The author launch an interesting study about MIT and GATT, which want to find the predictor factor of these MIGS proceduere. The details of the study, statistical methods, still need to be clarified before the article is published:

1) According to the author's narratives in limitation paragraphs, GATT is 360 degree trabcular meshwork cutting while MIT is a quadrant incision method. The blanching or staining difference between these two group should be analized in the manuscript. And the result of surgical success ( complete or qualified ) should also comparing between the two groups.

Answer: We certainly concur that a comparison of the blanching or staining method between the two approaches would be beneficial. Furthermore, it's important to highlight that we discovered no discernible difference in blanching patterns between GATT and MIT (the same reflected in the multivariate analysis with the type of surgery not defining the blanching or blue staining pattern). However, we wish to clarify that our intention was not to conduct a direct comparative study between GATT and MIT. Instead, our focus was on developing a simplified method for assessing the outflow pathway. A comparative analysis between these two surgeries, considering the variation in the number of quadrants operated, could introduce confounding factors. For instance, a greater extent of deroofing is logically expected to lead to a more substantial reduction in outflow resistance though this was not seen in this study. This consideration influenced our decision not to pursue a comparative study in this research. This information has already been detailed in the results section (outflow assessment subheading), indicating that a comparative analysis of this nature would yield similar outcomes. Yes, we are comparing the prospective clinical outcomes between GATT and MIT-we would be delighted to incorporate this suggestion to see the differences in blanching/blue staining patterns with other anterior segment imaging parameters in that study. 

2) The author use multivariiate regression module to analysis the predict facotr of surgical success. But the details table of this statistical analysis is not presented in the manuscript.

Answer: We wish to provide clarity regarding the inclusion of multivariate regression details and considered variables. These specifics have been outlined in the manuscript's Results section, specifically in the final paragraph. We are more than willing to share any additional information pertaining to this matter. Given that a significant association was not observed for most variables, we made the decision not to present this information in a table or figure and rather included the details in the text.

3) Whether the set position of the protractor is fixed, or based on the TM incision position (especially in MIT surgery), or the position where the irrigation needle is placed ? If it is only a partial incision (MIT), then the quadrants of blanching or staining will actually be affected by the incision site.

Answer: We want to provide further clarification regarding the protractor tool mentioned in our study. It's important to note that the protractor tool we utilized is a virtual online tool that is superimposed onto the image or video. It doesn't involve physically placing the tool on the eye during surgery. This tool is positioned over the eye in a manner that spans from limbus to limbus. Consequently, its placement is not influenced by factors such as the incision site or any similar considerations.

---

## [Editor Report · Decision Letter 2]

9 Oct 2023

‘Intraoperative predictors for clinical outcomes after microinvasive glaucoma surgery”

PONE-D-23-18067R2

Dear Dr. Rao,

We’re pleased to inform you that your manuscript has been judged scientifically suitable for publication and will be formally accepted for publication once it meets all outstanding technical requirements.

Kind regards,

Natasha Gautam, MBBS, MS

Academic Editor

PLOS ONE
---

## [Editor Report · Acceptance letter]

31 Oct 2023

PONE-D-23-18067R2 

*‘Intraoperative predictors for clinical outcomes after microinvasive glaucoma surgery”*

Dear Dr. Rao:

I'm pleased to inform you that your manuscript has been deemed suitable for publication in PLOS ONE. Congratulations! Your manuscript is now with our production department. 

Kind regards, 

on behalf of

Dr. Natasha Gautam 

Academic Editor

PLOS ONE